# A survey of current trends and suggested future directions in coral transplantation for reef restoration

**Sebastian C. A. Ferse** [1,2] *, **Margaux Y. Hein** [3,4], **Lena Rölfer** [1,2¤]

**1** Leibniz Centre for Tropical Marine Research (ZMT), Bremen, Germany, **2** Faculty of Biology & Chemistry (FB2), University of Bremen, Bremen, Germany, **3** Marine Ecosystem Restoration (MER) Research and Consulting, Monaco, **4** TropWATER, James Cook University, Townsville, Queensland, Australia

¤ Current address: Climate Service Center Germany (GERICS), Helmholtz-Zentrum Hereon, Hamburg, Germany
* sebastian.ferse@leibniz-zmt.de

**Data Availability Statement:** All relevant data are within the manuscript and its Supporting Information files. Data underlying all quantitative analyses are supplied as supporting information, S2 Table. This data has been anonymized by

## Abstract

Coral transplantation has been used in reef restoration for several decades, but information on the type of projects, their scope, scale, and success is mostly limited to published scientific studies and technical reports. Many practitioners do not have the capacity to share their progress in peer-reviewed literature, yet likely have a wealth of information to share on how to improve the efficiency of transplantation efforts. In order to incorporate non-published data on coral transplantation projects and gain an overview of the general features of these projects, we conducted an initial systematic online survey of projects run by various practitioners. Surveyed projects (*n* = 50) covered most of the tropical belt and ranged in size from a few hundred transplanted corals to >5000 transplants. The most frequent source of coral fragments were corals already broken from some previous impact ("corals of opportunity"; 58% of projects), followed by fragments stored in different types of aquaculture systems (42% of projects). The use of sexual reproduction was very limited. Fast-growing, branching corals were used in 96% of projects, being by far the most common transplanted growth form. About half of the projects mentioned undertaking maintenance of the transplantation plots. The majority of projects undertook subsequent monitoring (80%), yet the available data indicates that duration of monitoring efforts was not adequate to evaluate long-term success. The findings underline that while some general principles for successful coral restoration projects are reasonably well established, others need to be mainstreamed better in order to improve the effectiveness of coral transplantation for reef restoration. This relates in particular to sustainable funding, adequate site assessment, and long-term monitoring using established protocols. Additional information is needed to better understand and address potential challenges with regards to the sourcing of transplants and use of slow-growing species. A better integration of practitioners is necessary to improve the understanding of coral transplantation effectiveness. The results underline a need to develop and use monitoring protocols that allow gauging and comparing the effectiveness of coral transplantation among various projects, as well as for accessible platform(s) to allow the exchange of experiences made in different projects. Regular surveys of restoration projects

removing any information that may allow for identification of respondents, including editing of responses to open-ended questions.

**Funding:** SCAF acknowledges funding from the German Federal Ministry for Education and Research (BMBF, grant number 01LN1303A); https://www.bmbf.de/en/index.html. MYH is an employee of Marine Ecosystem Restoration (MER) Research and Consulting. The funders provided support in the form of salaries for authors SCAF (BMBF) and MYH (MER), but did not have any additional role in the study design, data collection and analysis, decision to publish, or preparation of the manuscript. The specific roles of these authors are articulated in the 'author contributions' section.

**Competing interests:** MYH is affiliated with Marine Ecosystem Restoration (MER) Research and Consulting. This does not alter our adherence to PLOS ONE policies on sharing data and materials.

are recommended to collate and share information among practitioners. We provide a number of recommendations for items to include in future surveys.

## Introduction

Coral reefs are the most diverse marine habitat, providing livelihoods to an estimated 500 million people, and are considered the ecosystem with the highest value per unit area on the planet in terms of the ecosystem services they provide [1,2]. In recent years, the drastic impact humans exert on reefs has become increasingly obvious, and almost two-thirds of all reefs are at immediate threat from local sources such as pollution, overfishing and destructive fishing [3]. Furthermore, the vast majority of the world's coral reefs is threatened by the effects of anthropogenic climate change [4]. The rapidly mounting anthropogenic threats to reefs have thus led to increasing calls for active management interventions, including reef restoration [5,6].

### Coral transplantation and reef restoration

The commonly used definition of ecological restoration is that of the Society for Ecological Restoration, which describes it as *"the process of assisting the recovery of an ecosystem that has been degraded, damaged, or destroyed"* [7, p.3]. An ecosystem is considered as successfully restored when it features sufficient biotic and abiotic resources to sustain itself structurally and functionally, attaining full recovery when all key ecosystem attributes (including absence of threats, species composition, community structure, physical conditions, ecosystem function, and external exchanges) resemble those of a reference system [8]. While restoration in a stricter sense aims at the re-establishment of pre-existing species composition and community structure, ecosystem rehabilitation emphasizes the reparation of ecosystem processes and services [7]. A recent definition of "restoration" for coral reef systems as *"any active intervention that aims to assist the recovery of reef structure, function, and key reef species in the face of rising climate and anthropogenic pressures, therefore promoting reef resilience and the sustainable delivery of reef ecosystem services"* [9, p.8] underlines a focus on ecosystem processes and services, and indicates that many reef restoration efforts at present are closer to the Society for Ecological Restoration's definition of ecosystem rehabilitation rather than restoration [7]. This may reflect the prevalent motivations underlying reef restoration efforts, but is also due to the fact that reef systems are rapidly changing, complicating the use of information on earlier community composition ("historical baselines") as reference for restoration [9,10]. Over the last few decades, a number of different methods have been developed, but coral transplantation, in which coral colonies and fragments are directly planted onto a reef substrate, is the most commonly used [11], and thus the focus of this research. Coral transplantation has been used in the restoration of coral reefs for a number of decades, particularly in cases with localized damage to reefs [12–15]. Other applications include the relocation of corals threatened e.g. by pollution or development [16–18] and transplantation in the frame of tourism activities [19,20]. More recently, the use of coral transplantation has been discussed with regards to conservation and reintroduction of endangered coral species [21] and the propagation of genotypes with increased resistance to adverse climatic conditions [22,23], the latter being an area of much current research interest and potential. Another common approach to restoration is "coral gardening" in which coral fragments undergo a "nursery phase", *in situ* or *ex situ*, prior to being planted on the reef [11,24,25]. Coral gardening complements coral transplantation,

providing an opportunity to further fragment and culture corals and thus maintain a repository of species and genotypes [24]. Advances in the propagation and rearing of corals from sexually-produced larvae are helping to increase genetic diversity and provide access to a large number of propagules for eventual outplanting, and in many cases have also reduced the need to obtain corals from the wild as source of transplants, particularly when collection of gametes is done *in situ* without the temporary removal of corals from the reef [26–28]. Yet, if coral transplantation and restoration in general are gaining interest and popularity as reef management strategies, the methods should ideally be used within a wider set of resilience-based, integrated frameworks (such as integrated coastal zone management [ICZM] or integrated water resources management [IWRM]) that simultaneously or pre-emptively address the source of threats and disturbances [9,20,29]. Restoration should be the last point of action in a carefully planned management framework [8,9,20]. Understanding the cause of coral mortality, the barriers to natural recovery, and the type of repair necessary to initiate recovery are all important considerations that need to be elucidated prior to undertaking transplantation [8,9,20,30].

In 1998, Alasdair Edwards and Susan Clark wrote a seminal article asking whether coral transplantation was a useful tool or, indeed, rather misguided meddling [30]. They reviewed the information on coral transplantation efforts conducted up to then and outlined a number of key recommendations with regards to transplantation. Importantly, they argued that coral transplantation is not warranted in reefs with sufficient levels of natural recruitment, and that transplantation should be considered a tool of last resort. Developments in the status of several coral species since then, as well as the plight of coral reefs in general, have increased the circumstances in which reef restoration, including transplantation of corals, is warranted to safeguard the integrity of reef habitats and ensure key functions are maintained in light of rapidly changing environmental conditions [5,31]. Yet, we posit that several main arguments of Edwards and Clark [30] remain valid. One key tenet of their study and subsequent guidelines on reef rehabilitation [20] is that the root causes of reef degradation need to be known and that, unless these causes are effectively addressed, recovery and success of transplantation are unlikely. However, while care is needed to ensure that the efforts spent on reef restoration are not in vain, incomplete information should not be seen as an excuse for inaction. Indeed, guidelines on reef restoration emphasize that coral transplantation is part of a portfolio of resilience-based approaches to address reef degradation that begins with proactive, no-regrets measures aimed at reducing negative impacts and increasing the recovery potential of degraded reefs such as reduced overfishing, improved water quality or enhanced local stewardship [9,29].

## Coral reef restoration: Science and practice

As in other fields of ecological restoration [7], the practice of reef restoration develops alongside, and sometimes parallel to, the science underpinning it (restoration ecology). While ideally, restoration practice and restoration ecology should draw upon and provide input to each other, in reality ecological studies may not be immediately applicable to restoration practice (e.g. because of discrepancies in scale), and valuable lessons to be learned from restoration projects may not be publicized. Furthermore, if developed ad hoc and without reference to, or awareness of, restoration guidelines, projects may not adhere to principles and standards of ecological restoration [e.g., 7,8,32].

In recent years, reef restoration or coral transplantation projects are increasingly adopted by the dive and tourism industry or as part of corporate social responsibility programs, raising the question to what extent the principles of good practice set forth e.g. by Edwards and Clark [30] and Edwards [20] or lessons learned from the restoration of other ecosystems [e.g.,

24,25,33] are implemented in current coral transplantation projects. Specifically, the questions arise whether causes of coral degradation are identified prior to the adoption of coral transplantation, whether there is a focus on fast-growing (usually branching) species to produce quick results, and whether adequate follow-up monitoring of transplanted corals is carried out. Furthermore, if projects are conducted by entities without sufficient links to regulatory agencies (which may be the case e.g. for tourism operators), they might not be accompanied by additional management measures addressing ongoing stressors to degraded reefs.

Until recently, systematic reviews of coral transplantation projects were missing from the scientific literature, but the last five years saw a number of important reviews of coral reef restoration studies. Some reviews focused on cost-efficiency of specific methods [34,35], or on indicators of social-ecological effectiveness [36]. Other reviews assessed the types of methods used [37], and the extent to which other key ecological processes (e.g. competition, herbivory, nutrient cycling) are considered in coral restoration efforts [33,38]. Lirman and Schopmeyer [37] concluded that the use of coral nurseries is a preferred method in restoration projects nowadays, limiting the amount of transplants collected from wild stocks. However, as these reviews were limited to published scientific studies, they are likely to miss projects by the non-academic sector, potentially leading to a bias in terms of duration, scale and objectives [35]. A notable exception is the review by Young et al. [39], who conducted a thorough survey of restoration projects and practitioners. However, it was limited in terms of species (only the genus *Acropora* was covered) and geographic coverage (Caribbean). More recently, Boström-Einarsson et al. [11] provided the most comprehensive review of restoration studies to date, including a survey of practitioners. That review, though, did not systematically differentiate between scientific studies, the grey literature, and practitioners, and stopped short of assessing differences between projects by different actors and with varying objectives. To address this gap, we developed an online survey targeting practitioners from a range of backgrounds involved in coral transplantation projects. The purpose of this study was the assessment of coral transplantation projects in different parts of the world, particularly ones not reported in the scientific literature, to provide an overview of current practices and actors involved in reef restoration projects. Specifically, we investigated i) to what extent restoration projects by different types of practitioners differed in terms of objectives and methods; ii) whether there continues to be a focus on faster-growing, branching species; iii) whether assessments of the causes of reef degradation are regularly included in projects by the non-academic sector; and vi) the source of corals for transplantation.

## Materials and methods

A voluntary self-reporting online questionnaire consisting of seventeen questions (dichotomous, multiple choice, open-ended, see S1 Appendix) was distributed between September 2016 and August 2017. Specifically, questions focused on the objectives of the transplantation project(s), the number, type and source of transplanted corals, and what kind of pre- and post-transplantation monitoring and maintenance was carried out. Furthermore, information on the location of the project, the type of organization carrying out the project (i.e. government, NGO, business, private or academic), the source(s) of funding, and additional measures accompanying transplantation were collected. Organizations conducting coral transplantation were searched online using the keywords CORAL RESTORATION, REEF RESTORATION and CORAL TRANSPLANTATION in Google and approached via email using contact information provided online. "Transplantation" was considered broadly as any process that involved the relocation of coral fragments or colonies to a recipient site in the reef, irrespective of the source of transplants. A total of 43 potential respondents were identified in this process,

12 of which responded to our request. More organizations were recruited through existing personal contacts and referrals from previous respondents. In addition, a message addressing potential participants was posted on the Coral-List (http://coral.aoml.noaa.gov/mailman/listinfo/coral-list), a moderated mailing list with >9000 subscribers, including practitioners with both academic and non-academic backgrounds, and hosted by the US National Oceanic and Atmospheric Administration (NOAA). The contacted persons were given information about the study and the link to the survey. The survey followed the principle of prior informed consent: all participants were informed about the background and aim of the survey, and consent was assumed implicitly when respondents proceeded to fill the survey. Participation in the survey was voluntary, and the framing and setting of the survey was non-coercive. The nature of information collected poses minimum harm to participants and thus did not require written consent. The collected information was treated confidentially, and data are made available only in anonymized form. Respondents were asked to choose what best describes their organization, i.e. 'government', 'NGO', 'business', 'private', 'research institute/university' or 'other'. For coral growth forms, the categories 'branching', 'massive' and 'other' were offered. Where individual organizations were involved in multiple coral transplantation projects, they were asked to fill separate questionnaires for each project. The acquired data was analyzed using the 'Deducer' package in R [40]. Open-ended questions were coded into categories and then further analyzed. For nominal data, differences in the frequency of answers among categories were assessed using a Chi-Squared Test; in cases where cells contained zeros, a Likelihood Ratio (G) Test was applied instead. In the case of 2x2 cross tables, a Fisher's Exact Test was performed. The last question, which addressed lessons learned and allowed for a broad range of responses, was analysed using the program NVivo (Version 11.4.2 (2018)).

## Results

### Size and nature of coral transplantation project

A total of 43 respondents involved in transplantation projects participated in the survey, providing responses for a total of 50 projects (S1 Table). Respondents were mainly from NGOs and academic institutions, and were distributed among developing and industrialized countries from around the (sub-)tropics. 21 projects were located in the Caribbean, 28 in the Indo-Pacific, and one in the Mediterranean (Fig 1, S2 Table). As respondents were asked to self-identify which kind of organization they belonged to, there was some inconsistency in the responses, which was adjusted prior to further analyses. One university self-identified as 'government' but was included with 'Research institute/university' for consistency. The distinction between 'private' and 'business' was not always clear. Overall, the private/business category included one company conducting large-scale transplantation related to dredging and construction activities (2 projects) and a commercial coral farm. The remainder were researchers/consultancies working on reef restoration and conservation ($n = 4$) and dive resorts or associated education centers ($n = 6$). To be able to assess whether tourism-associated projects differed from those by others in the private sector, such as restoration consultancies, the self-identified 'private' and 'business' categories were reassigned into 'tourism' and 'other private sector' projects.

**Scale and type of coral restoration projects.** Most projects (82%) were either small- or large scale (<500 or >5000 corals transplanted, respectively), with only a single project (NGO from Malaysia) using between 500 and 1000 corals transplants. For the remaining analyses, this project was grouped with the projects using <500 transplanted corals. The scale of the projects did not differ significantly among the types of actors or regions (Likelihood Ratio, n. s.; Fig 1, Table 1).

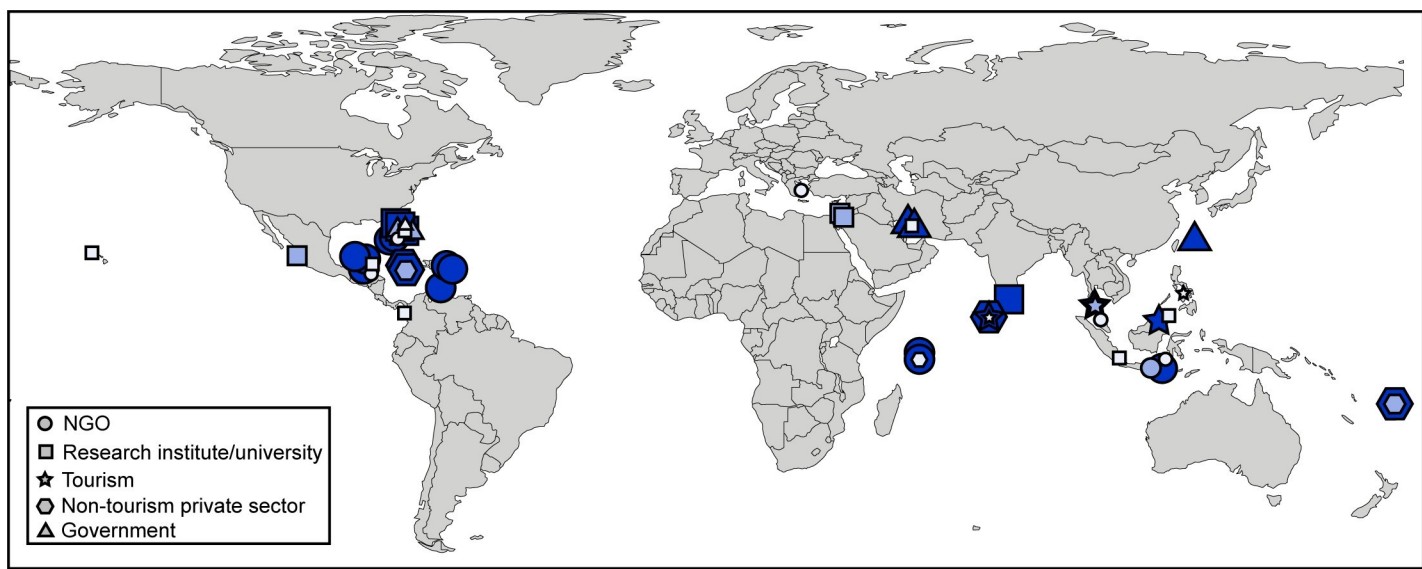

**Fig 1. Map showing the 50 coral transplantation projects covered in this survey.** Symbols represent the different types of organizations running the projects, and size of the symbols corresponds to the number of corals transplanted in each project (small/light blue: <500 corals, intermediate/medium blue: 1000–5000 corals, large/dark blue: >5000 corals). The NGO project in Malaysia used between 500 and 1000 fragments, but was grouped with the <500 corals projects for further analysis. Made with Natural Earth.

In most cases, transplantation projects were funded through government sources, followed by private and NGO funding (Fig 2). Note that while respondents could list several types of sources of funding, it is possible that indirect contributions of particular funders may not have been captured and fully resolved; for example, NGOs usually receive donations from private and business sources or support in the form of government funding, which they may then use to fund restoration projects.

**Main objectives of coral restoration projects.** 'Habitat restoration' was the most commonly stated objective for restoration (given for 80% of projects), and 'research' was the least selected objective (6%) (Fig 3). Only the objective 'creation of a tourist attraction' differed significantly among types of respondents, being mentioned by 83% of tourism projects, 43% of the other private sector and 41% of the NGO projects, but only by 21% of the academic and none of the government projects (Likelihood Ratio, G = 12.791, df = 4, $p$ = 0.012). A trend (albeit non-significant) was also visible for the objective 'research', which was stated as the main objective for 33% of the government projects, but only for 7% of the academic and none of the other projects (Likelihood Ratio, G = 7.854, df = 4, $p$ = 0.097).

**Table 1. Number of coral transplantation projects for different types of respondents, and scale of projects (number of transplanted corals).**

| Scale | Type of respondent | | | | | Total |
|---|---|---|---|---|---|---|
| | Government[1] | NGO | Tourism | Other private sector | Research institute/university | |
| <500 | 1 | 5 | 2 | 1 | 7 | 16 |
| 1000–5000 | 2 | 1 | 1 | 2 | 3 | 9 |
| >5000 | 3 | 11 | 3 | 4 | 4 | 25 |
| Total | 6 | 17 | 6 | 7 | 14 | 50 |

Row and column totals are given to the right and below.

[1]One university self-identified as 'government' but was included with 'Research institute/university' for consistency.

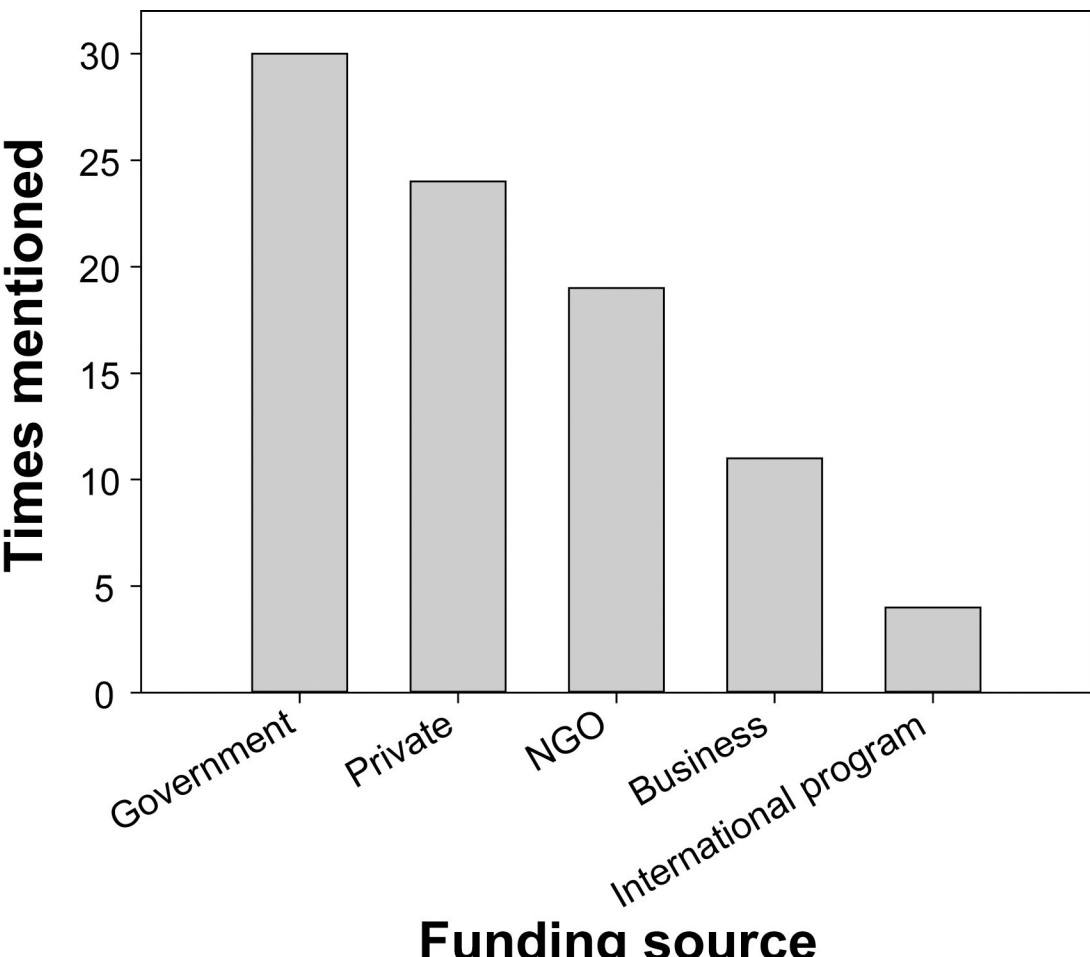

**Fig 2. Funding sources for the transplantation projects.** Multiple responses were possible for each project.

**Type and source of coral transplants.** 'Branching' corals were transplanted most frequently, with 'massive' and 'other' growth forms being less common (96%, 54% and 28% of all projects, respectively). No differences in the frequency of use of any of the growth forms existed between Caribbean and Indo-Pacific projects. There were significant differences among actors in the use of 'other' growth forms (Chi-Squared Test, $\chi^2 = 9.908$, df = 4, $p = 0.042$), which were mostly used by the private sector (71%), followed by government and tourism projects (both 33%), NGOs (23%), and academic projects (only 7%).

As source of coral transplants, 60% of all projects used already-broken fragments ("corals of opportunity"), while 46% of projects also sourced fragments from colonies in the reef. 42% of all projects used some form of aquaculture (including *in situ* or *ex situ* nurseries) to produce corals to transplant, while 8% stated 'other' as source of transplants. Only three projects (6%) explicitly mentioned sexual reproduction of corals. While the use of "corals of opportunity" did not differ significantly among types of respondents, a trend was visible in the targeted harvest of corals for transplantation from colonies in the reef (Chi-Squared Test, $\chi^2 = 9.410$, df = 4, $p = 0.052$). It was most common for projects from the private sector (86%), followed by government projects (67%), academic projects (50%), and tourism and NGO projects (33% and 24% respectively).

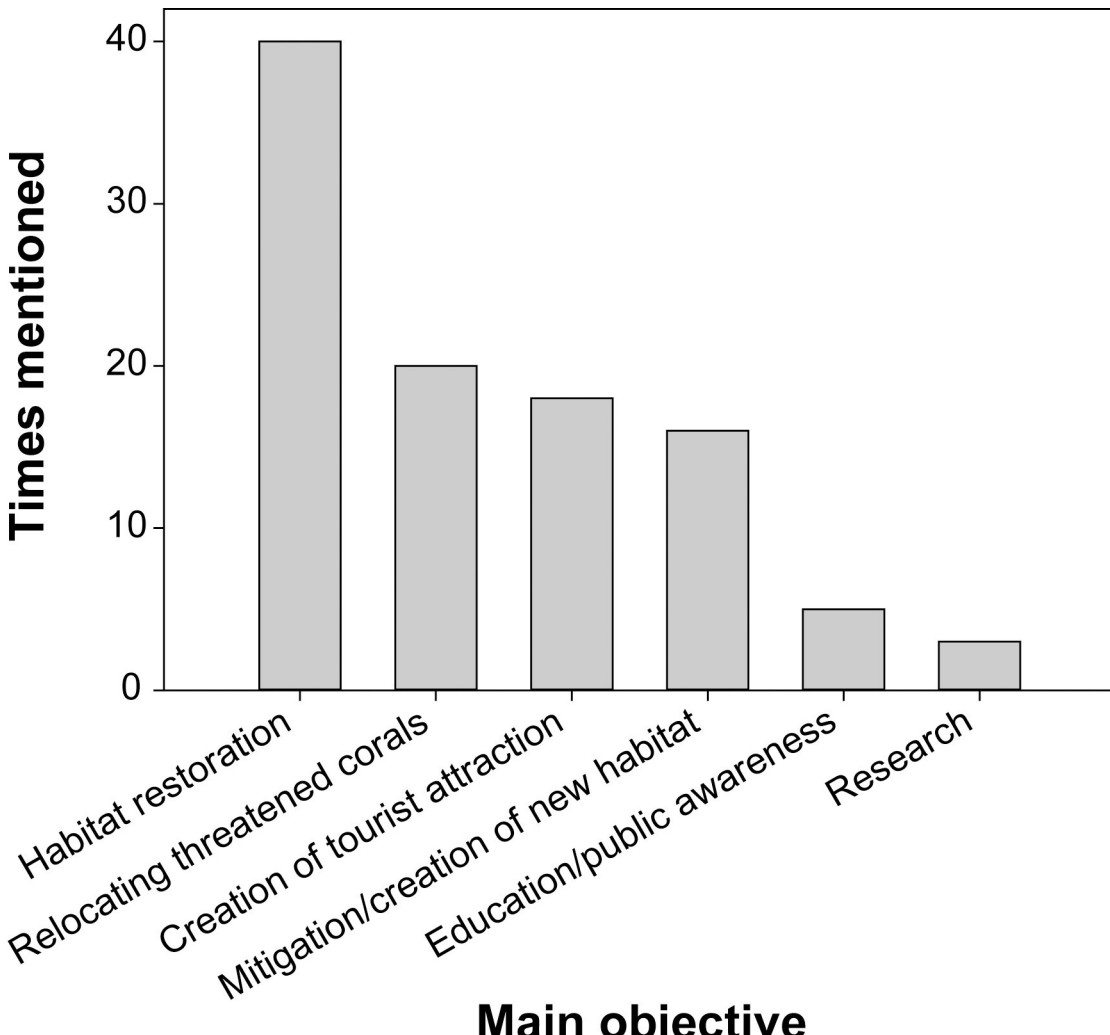

**Fig 3. Overall main objectives of the different transplantation projects.** Respondents were asked to provide multiple answers if several objectives were deemed equally important.

Aquaculture as a source of transplants did not differ significantly among types of respondents, but according to region. 13 out of 21 Caribbean projects used aquaculture, while only 8 out of 28 Indo-Pacific projects sourced transplants from aquaculture (Fisher's Exact Test, $p = 0.04$). Aquaculture was also significantly more prevalent in larger-scale projects. Only 13% of the small projects (<500 corals transplants) sourced corals from aquaculture, while 67% of the medium and 52% of the large-scale projects did (Chi-Squared Test, $\chi^2 = 8.990$, df = 2, $p = 0.011$). Other sources of transplants did not differ significantly with scale of projects.

In the majority of projects (52%), corals were transplanted onto natural reef substrate only, while in eight projects (16%) corals were transplanted exclusively onto artificial structures. In 32% of projects, corals were transplanted onto both artificial and natural substrates. Artificial substrates alone were most common in the small-scale projects (six out of 16), while a mix of substrates was significantly more common in large-scale projects (eleven out of 25; Likelihood Ratio, G = 11.372, df = 4, $p = 0.023$). Projects aimed at the creation of tourist attractions were more frequently transplanting onto artificial or a combination of artificial and natural

substrates (50% and 56% of those were aimed at tourism, respectively) than onto natural reef substrates (only five of 26 doing so were for tourism purposes; Chi-Squared Test, $\chi^2$ = 6.702, df = 2, $p$ = 0.035).

## Monitoring, assessment and maintenance

Assessment of the causes of reef degradation was missing in 32% of all projects. It was included significantly more frequently in large-scale projects (84%) than in small-scale projects (50%; Likelihood Ratio, G = 6.157, df = 2, $p$ = 0.046). However, whether assessment of the cause of degradation was carried out or not did not differ among the types of organizations. The single most common assessment was some form of monitoring, while anecdotal or secondary information together were used to gauge the cause of degradation nearly half of the time (Fig 4A). Choice of transplantation substrate was not significantly different between projects with and without prior assessment of the cause of degradation.

Environmental conditions at the recipient sites were also assessed in most projects (78%). Again, whether environmental conditions were assessed or not did not differ among types of respondents. Monitoring was by far the most common form of assessment (Fig 4B). Yet, none of the respondents mentioned assessment of coral recruitment at the recipient site.

Most projects ($n$ = 47) conducted further work following transplantation. This included maintenance such as cleaning (e.g. removal of algae, sponges and coral predators; 36% of the projects) and repairs (e.g. reattachment of loose fragments, fixing artificial structures; 26% of the projects), as well as follow-up monitoring (80% of all projects). All three projects not mentioning any further work were carried out by non-tourism private sector actors, and whether further work was conducted thus differed significantly between these actors (only 57% of which did) and all other actors (all of which carried out some kind of further work; Likelihood Ratio, G = 13.136, df = 4, $p$ = 0.011). Individual types of further work (follow-up monitoring, cleaning and repairs) did not differ significantly among types of respondents. While we did not specifically ask for the length of follow-up monitoring, for five projects (10%) respondents stated to have carried out monitoring for four years or longer, and two of those were

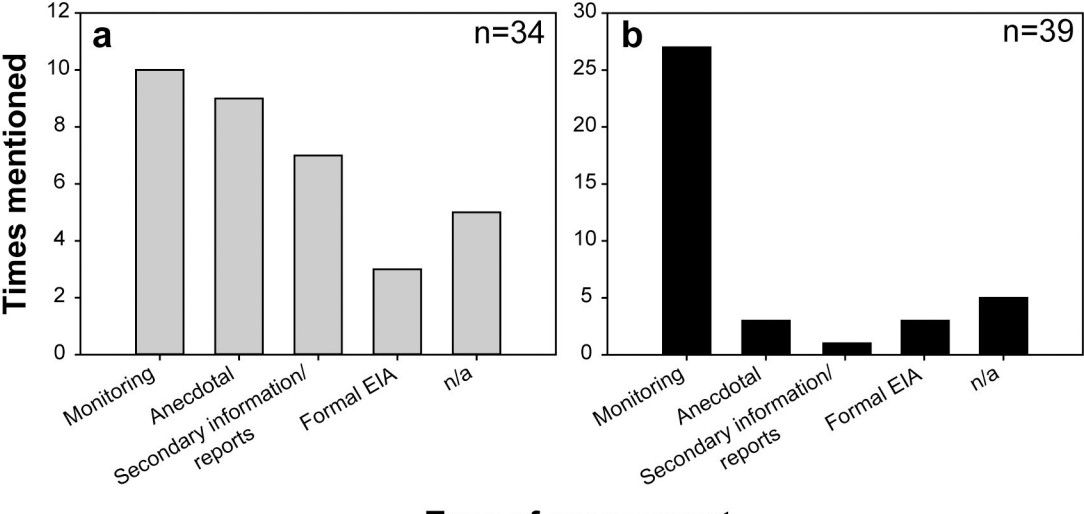

**Fig 4. Sources of information used in projects.** The types of assessments and sources of information used to gauge a) the cause of coral degradation prior to transplantation of corals, and b) environmental conditions at the target site for transplantation.

monitoring for >10 years. In six projects monitoring lasted for 2 years or less, and another five conducted 'regular' monitoring every six or 12 months for unspecified lengths of time.

## Measures surrounding transplantation

A range of additional measures were conducted in the different projects (Fig 5). The most common was 'education and public awareness'. Local management actions were less common, and only a few projects included concrete measures to improve environmental conditions. There were some differences among actors in terms of additional measures surrounding the projects. No additional measures were included in 50% of government and 21% of the academic projects, while 86% of projects by other private sector actors and all of the tourism and NGO projects were accompanied by further measures besides coral transplantation (Likelihood Ratio, G = 11.889, df = 4, $p$ = 0.018). A trend was visible in the use of fishing regulations, which accompanied 53% of NGO, 50% of tourism and 43% of non-tourism private sector projects, but only 7% and 17% of academic and government projects, respectively (Chi-Squared Test, $\chi^2$ = 8.951, df = 4, $p$ = 0.062).

## Lessons learned

For a total of 30 projects, respondents mentioned a number of considerations that according to their experience would maximize the success of restoration efforts (S3 Table). Most comments related to methodological, technical and design aspects, but a few observations were also offered regarding public outreach, staff management, environmental impacts and integration of additional management measures. For example, the high cost of transplantation and need for sufficient, sustainable funding was most frequently mentioned ($n$ = 7 respondents).

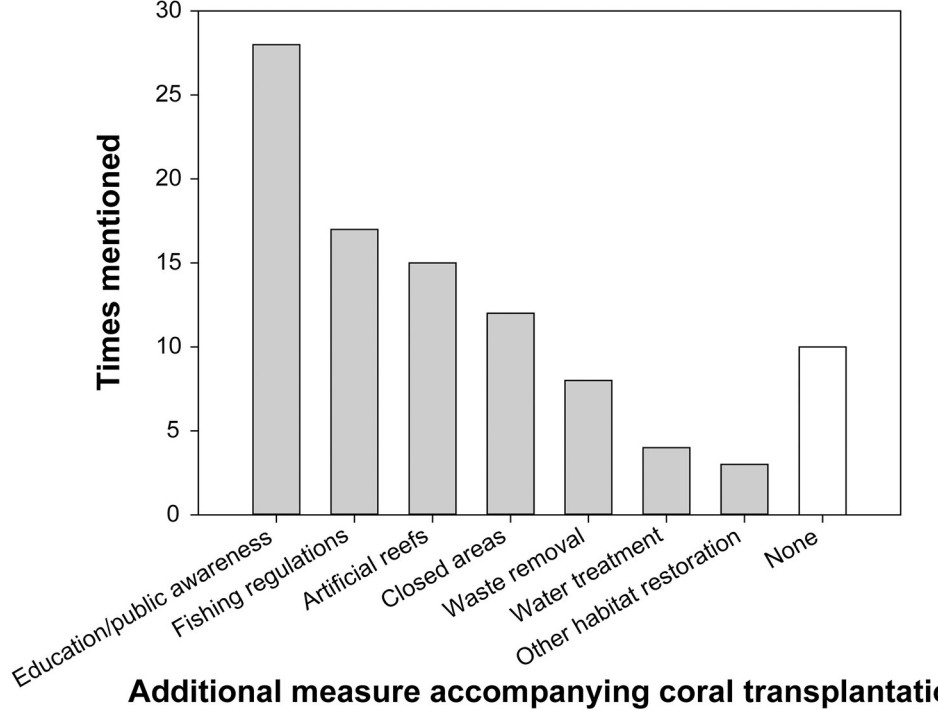

**Fig 5. Measures in addition to coral transplantation carried out in the different projects.** Multiple responses were possible for each project.

Interestingly, several respondents pointed out that even if transplantation efforts were not able to meaningfully restore larger areas of reef, the increased awareness reached by involving local stakeholders or tourists was beneficial for reefs by itself. Public involvement was the second-most mentioned theme. Also mentioned multiple times was the importance of regular monitoring, maintenance, and appropriate environmental conditions at the recipient site. Importantly, several comments underlined the role of understanding and emulating natural processes and conditions, such as diversity in transplanted species, adequate spacing, and prevalence of territorial algal-farming damselfish in recipient sites. The need for an adequate legal framework was pointed out several times; one respondent mentioned their project was able to produce corals in *ex situ* aquaculture faster than permits were available for transplanting them to the reef. Finally, a number of respondents pointed out that restoration was unlikely to succeed unless adverse environmental conditions (e.g. pollution, overfishing) are addressed, and that large-scale environmental stress (e.g. thermal anomalies and storms) can negate restoration efforts.

## Discussion

In this review, we provide an overview of coral transplantation projects from around the world beyond those reported in the scientific literature, including the private sector for which little information exists to date, and identify differences in the size and objective of projects and in accompanying measures among different types of practitioners. Being able to draw on the knowledge gained by projects not reported in the scientific literature is indispensable to obtain a better overview of current techniques and to tailor future recommendations. The present survey constitutes a first step in that regard, but regular, more detailed surveys among practitioners and sharing of structured information among a community of practice are strongly recommended. The results underline that while advances in coral restoration have been made over the past decades and some principles are adopted in most projects, several shortcomings of transplantation projects identified two decades ago are still wide-spread. Furthermore, the results of our survey show some aspects that contrast with the findings of several recent reviews predominantly drawing on the scientific literature. Interestingly, a number of differences were found among the types of respondents, underlining that the approaches and objectives of reef restoration projects vary depending on who is running them.

### Geographic distribution of transplantation projects

The majority of projects surveyed were from the Caribbean, particularly Florida. This reflects the historical development of reef restoration, for which a lot of pioneering work took place in that area [see 41]. While this survey should not be seen as exhaustive, a notable paucity of projects existed for Australia, where the recent reports of degradation of the Great Barrier Reef have fuelled interest in active restoration efforts [5]. The Australian government is currently placing a lot of emphasis on reef restoration following the dramatic reports on the effects of the 2016–2018 bleaching event on the Great Barrier Reef (GBR), having recently allocated 100 million AUD towards reef restoration and adaptation [34], and more projects are likely to develop there in the near future. In June 2020, a first "Restoration Hub" was created by The Great Barrier Reef Foundation to assist further implementation of coral reef efforts and facilitate transfer of knowledge, resources, and experience across Traditional Owners, scientists, tourism operators and the local community on the GBR [42]. Tourism operators in that region are highly motivated to further engage in coral reef restoration efforts to improve coral cover and resilience, but also increase education and public awareness on the threats to coral reefs [43]. The recent review by Boström-Einarsson et al. [11] includes 16 case studies from

Australia, the majority of which are from 2017 or later. Generally, an increasing interest in reef restoration is notable both in terms of number of published studies [11,37,38] and in the wide geographic range of restoration projects reflected in our results. In the current era of ubiquitous anthropogenic impacts on coral reefs, active reef restoration is increasingly seen as a necessary approach to sustain the ecological functions and services provided by these ecosystems [5,44]. The majority of member countries of the International Coral Reef Initiative (ICRI) are either already involved in, or interested in getting involved in, coral restoration, and are calling for new policies and funding schemes specific to coral reef restoration [45]. Government funds (including intergovernmental agencies such as the World Bank or GEF) were the largest category for funding (supporting 60% of the projects), and a number of recent international initiatives emphasize restoration (e.g. the 2021–2030 UN Decade on Ecosystem Restoration) [9], which further indicates that reef restoration is increasingly placed on government agendas. In such a context, we expect a further uptake of coral reef restoration as a management strategy in most reef regions of the world in the next decade.

## Objectives of transplantation projects and transplanted corals

While habitat restoration was listed as the most frequent objective of the surveyed projects, the survey did not ask for specific goals and whether these were defined beforehand. This point should be a key item in future surveys, as definition of specific targets and goals is not only an important principle of ecological restoration [8,9,46] but also allows for assessing whether projects are on track for reaching them, and thus should inform accompanying monitoring. 'Research' was identified as main objective in three out of the 50 projects, with only a single project specifying it as the only objective. While that project thus should not be considered an actual reef restoration project (i.e. it is an example of restoration ecology, rather than ecological restoration), the large majority of the surveyed projects would qualify as such.

Branching corals were by far the most common growth form transplanted. Massive corals, which grow slower but may survive transplantation much better [14], were used in half of the projects. Even though the use of massive or other slow-growing species has considerably increased over the past decade through the development of micro-fragmentation techniques [47], the remaining focus on branching species may to some extent reflect differences in objectives and regional specifics. As coral transplantation projects in the Caribbean focus heavily on branching acroporid species [37,39], the fact that 42% of the projects were from the Caribbean might be expected to lead to an overrepresentation of branching corals. However, there were no differences in the frequency of using particular growth forms among regions. Furthermore, only one Caribbean project had the explicit objective of increasing a threatened coral population, while one academic project from the Indo-Pacific was aimed at research into branching coral growth rates. The emphasis on branching species in the other projects thus seem to reflect a prevailing strong focus on fast-growing branching species in transplantation projects around the world [38]. Other growth forms were still less common, underlining that transplantation in most projects did not establish diverse coral communities. The recent review by Boström-Einarsson and colleagues found an even stronger focus on branching species [11]. Using slow-growing species that may be more resilient to threats could improve long-term success of restoration efforts. However, the reasons for choosing fast- over slow-growing species can be manifold, including: availability (i.e. branching species are more likely to naturally fragment), lack of permits (i.e. some restoration efforts are specifically permitted for endangered *Acropora* sp.), technicalities (i.e. collection of slow-growing colonies is more damaging to the reef than from branching species, and fast growth of branching species provides faster results), or a deliberate strategy (e.g. initially using fast-growing species to create structure,

followed by transplantation of slower-growing species later on). While techniques for propagating and outplanting of massive species have considerably advanced within the last decade (e.g., [47]), it may take more time for them to become mainstreamed. Furthermore, timing, funding and a lack of permits for the collection of massive species may cause obstacles to simultaneously transplant other growth forms in addition to branching corals. Acknowledging these limitations, re-establishing diverse communities would provide more niches for reef-associated biota [e.g., 48,49] and a higher range of potential responses to stressors such as thermal anomalies or diseases [50]. Transplanting corals in mixed-species plots may also be used to reduce corallivory and increase coral growth rates [33]. Thus, while the simplicity of the survey questions may not have allowed the projects to fully explain why only branching corals were used, it appears that there is still considerable scope for mainstreaming the use of non-branching coral species in restoration projects, including in the Caribbean. To better assess this, future surveys should explore the rationales for choice of transplanted species in more detail.

Mariculture and nursery production of coral fragments, which is costly and work-intensive but can significantly reduce damage to the reef and survival of transplants [51,52], was found in almost half of the projects, but collection of fragments from the reef was still more widespread, particularly among the smallest projects. The tendency of government and non-tourism private-sector projects to more frequently source corals from the wild may reflect the nature of these projects; e.g., salvaging and mitigation projects are most likely to be run by these kinds of actors rather than by NGOs or the tourism sector. The relatively low prevalence of mariculture and nurseries as source of transplants observed here is in contrast with the results of recent reviews of the scientific literature, which concluded that the spread of coral nurseries is replacing wild stocks as the preferred source of coral transplants [37,39]. Boström-Einarsson et al. [11] reported coral gardening in 48% of surveyed projects, compared to only 20% using direct transplantation. Furthermore, mariculture/nurseries as source of transplants was more widespread in the Caribbean, which was the focal area for the reviews by Lirman and Schopmeyer [37] and Young et al. [39]. Many projects in this region have been operating on longer time scales, and the availability of wild coral stocks is much lower than for other parts of the world, with Caribbean *Acropora* species listed as critically endangered by the IUCN. One explanation for the apparent difference in projects harvesting directly from the reef may be that coral gardening includes the use of fragments taken from the reef and grown in nurseries for some period of time. Since we specifically asked for 'aquaculture' as source of transplants, as opposed to the harvesting of corals from the reef, our category was more conservative, and several projects classified under collection of corals from the reef in our survey might have been categorized as coral gardening using the classification e.g. of Boström-Einarsson et al. [11]. Coral transplants produced using sexual reproduction, on which significant advances in research have been made over the past decade [53], were still very rare in the reviewed projects, which reflects the results of other reviews. Overall, our results indicate that the use of asexually and sexually reproduced corals for transplantation remains yet to be mainstreamed in restoration projects, in particular beyond the academic sector.

## Assessment of reef condition

Most projects did not undertake environmental assessments prior to transplantation. In particular, no project mentioned assessing patterns of coral recruitment, and only 1/3 of the projects surveyed assessed initial causes of reef degradation. This is worrying, given the core concept of ecological restoration that assisting natural recovery processes is the most reliable way to achieve recovery [8]. The lack of such assessments potentially leads to poorly planned

projects or a choice of inadequate approaches [8,20]. For example, coral transplantation may be a waste of time and resources unless the recipient site fails to recruit juvenile corals [30]. Furthermore, the absence or minimization of threats is a key attribute of ecosystems that are to be restored [8]–transplantation is not likely to succeed unless chronic stressors are removed [20,54]. Individual genotypes or populations that survive episodic stressors, either due to intrinsic resistance or locally-favorable environmental conditions, may provide recruitment in the meantime and allow a buildup of resilience while chronic stressors are handled [55]. However, if adult coral populations are diminished and/or stressed to the extent that reproduction is compromised, transplantation of adult corals in addition to ensuring minimum adequate conditions for coral survival and reproduction may be needed to restore successful recruitment of juvenile corals. Information on stress-resistant genotypes and environmental conditions can provide guidance on particularly promising candidates for transplantation [56].

An in-depth understanding of the causes of degradation is also key to developing goals and objectives for a restoration project and optimizing site-selection so that the project has a chance to withstand future disturbances [46]. As increasing information on previous successes and failures becomes available, valuable lessons for selecting suitable sites for multi-species restoration can be drawn [e.g. 57, p.230]. Although an assessment of the environmental conditions at the transplantation sites was done in some of the projects surveyed, many respondents relied on second-hand and anecdotal information in assessing their project sites. However, this does not necessarily demonstrate inadequate preparation–dedicated surveys of environmental conditions, and in particular of coral recruitment, are costly and time-consuming, and relevant information may be gathered from individuals familiar with the local environment or from public sources. For example, remote sensing data on particulate matter or sea surface temperature fluctuations can help to identify sites with higher potential for transplant survival and resilience to future disturbance [58,59]. Lastly, while the need for baseline surveys prior to restoration is highly relevant [60] and better management and water quality standards remain important, they should not be considered excuses for foregoing attempts at saving remaining diversity and resilience.

## Accompanying measures

Tools of reef restoration such as coral transplantation should form part of a larger strategy aimed at increasing ecosystem resilience and assisting the recovery of a degraded, damaged or destroyed ecosystem. Reef restoration itself complements other conservation activities and nature-based solutions (e.g. [61]), and vice versa [8]. While Edwards and Clarke [30] noted that coral transplantation should constitute a tool of "last resort", developments both in the state of reefs and restoration ecology over the past two decades mean that transplantation of corals is developing into a central piece of the reef restoration toolbox. For example, according to Young et al. [39], 90% of restoration practitioners in the Caribbean believe that *Acropora* populations there may not recover without active intervention, in particular coral transplantation. The risk remains that transplantation of corals is becoming a "quick fix" to fight symptoms rather than causes of reef degradation and is applied in isolation. Allaying this risk, our finding that the large majority of transplantation projects included additional measures indicates that a multipronged approach encompassing various tools is common nowadays. Nearly all projects included some form of follow-up work, with the exception of a few projects by the non-tourism private sector. This may reflect differences in objectives, e.g. translocation of corals in the frame of construction activities, as well as in project duration. Over half of the projects included some form of education and public awareness. As one respondent put it, "the key success factor is the people" (S1 Table). Hein et al. [36] argue that stewardship and capacity

building are two key indicators for social-ecological success of coral restoration. Yet, involvement of local stakeholders was not ubiquitous, and we suggest it should be made an integral part of any reef restoration effort. Besides increasing local support and sustainability of restoration projects [62,63], public awareness can assist in raising funds for restoration projects, which could address in particular the issue of long-term monitoring and maintenance. The importance of removing chronic stressors at the restored site in order for transplantation to be successful [64] also appears to be increasingly recognized, even if dedicated monitoring of environmental conditions was lacking in many cases. Several of the projects remarked on measures to address environmental conditions, such as wastewater treatment, fishing regulations or closed areas, and respondents mentioned working with other stakeholders (e.g. fishers or local community members) to achieve these. These responses underline that collaboration among different stakeholders may be necessary to ensure amenable conditions for successful restoration [65]. Our survey was unable to ascertain the presence of chronic stressors or the original cause of reef degradation, and we thus cannot assess to what extent root causes of degradation or chronic stressors were adequately recognized and addressed in each case. Thorough, systematic documentation of environmental conditions and explicit statement of goals is an important component to tailor restoration projects to local conditions and gauge their success [32], and should be assessed more specifically in future surveys.

Interestingly, there was a difference among actors in additional measures that was contrary to our initial assumptions. Private sector (including tourism) and NGO projects more frequently employed additional measures than government and academic projects, and fishing regulations also tended to follow this division. Entrepreneurial engagement in marine conservation, such as protected areas established by the private sector, is receiving increasing attention in recent years [66]. A noteworthy example of a coral transplantation project in terms of area covered (>2ha), cost effectiveness, enhanced coral cover, high species diversity and coral survival was initiated by a private Corporate Social Responsibility initiative [67]. Such projects may be at an advantage in terms of available funding (i.e. long-term financial sustainability), local social capital (but see [68]) and higher institutional flexibility (i.e. less 'red tape') than government-run initiatives [66], making it easier to include additional measures such as community engagement, livelihood diversification or environmental awareness campaigns. At the same time, such projects may be subject to less oversight and accountability [69], potentially jeopardizing restoration success [62]. On the other hand, academic and government projects may have narrower, specifically-focused mandates, focusing only on the technical aspects of coral transplantation and reef restoration, while additional measures are treated separately. Yet, entrepreneurial conservation programs are subject to market forces and may thus falter, and state support as well as adequate engagement of local stakeholders is necessary to create durable, sustainable institutional arrangements [62,68,70].

## Collecting and exchanging experiences

Many respondents seemed to learn much from their own experiences, as shown by the additional comments. For example, one respondent reported higher survival and growth for smaller fragments of *Pocillopora damicornis* compared to larger ones, underlining that size and survival are not necessarily correlated in all species [30]. A project from Malaysia pointed out negative effects of algae-farming damselfish on transplant survival, which have so far been mainly described for Caribbean projects, while restoration efforts in the Indo-Pacific were assumed to benefit from territorial damselfishes as they provide protection from corallivorous fishes [33]. There currently is a paucity of platforms on which to document and share such experiences made in reef restoration. This need has also recently been pointed out by others

trying to obtain an overview of existing reef restoration projects involving coral transplantation [71]. The recently-formed Coral Restoration Consortium (CRC; http://crc.reefresilience.org/), which among other objectives includes the establishment of a database of coral nurseries and restoration sites, may provide such a platform. The newly-launched interactive database developed by Boström-Einarsson et al. [11] constitutes a much-needed resource summarizing information from the scientific and grey literature as well as several dozen reef restoration projects.

Furthermore, while monitoring and maintenance are identified as key points of coral transplantation efforts [20], the lack of monitoring standards and guidelines has thus far impeded the measurement of success across social and ecological scales. Here, most of the projects (80%) mentioned post-transplantation monitoring. However, the time scales rarely appeared sufficient to adequately judge the performance over ecologically meaningful periods in all but a few cases, mirroring the findings of Bayraktarov et al. [35] for scientific studies. While few differences in monitoring among types of actors were observed in the present survey, there were differences with regard to additional measures. These may have been related to permitting and funding requirements (which can be very strict, e.g. in the case of coral gardening in the U.S. [37]), and it would be desirable that consistent minimum standards of accountability and monitoring apply to any reef restoration project. Developing monitoring guidelines over meaningful timeframes has been the focus of a number of studies in the recent years [36,60]. The Coral Reef Restoration Monitoring Guide recently developed by the CRC [60] presents a number of metrics that should be consistently monitored across all projects to improve standardisation and the ability to compare effectiveness across regions, and methods. Other metrics may also be used depending on the specific goals of the project, for example by incorporating reef-scale ecological metrics, as well as socio-economic metrics [60]. Importantly, monitoring should also include control sites, particularly reference ecosystems, or reference objectives in order to track success through time [8,20,36], which furthers highlights the need to monitor environmental conditions prior to restoration. Given the current rate of change on reefs around the world, the reefs to restore constitute 'moving targets', and using information on the earlier condition of a site does not provide adequate reference for restoration since the composition of viable benthic assemblages will be different under future environmental conditions [5,9]. Restoration targets should thus include the restoration of vital ecosystem processes rather than particular status indicators such as percent cover by a particular species [33,36]. A similarly critical issue is that specific goals and objectives, which are needed to gauge the long term success of restoration projects and should thus be stated in advance and ideally be discussed with all concerned stakeholders [8,63], are rarely defined in reef restoration projects [35,36]. In part, this may be related to a lack of information on the initial cause of degradation, observed in a third of the surveyed projects, and the difficulty of defining key target ecosystem attributes as a basis for objectives and goals [8]. The new CRC monitoring guidelines, which take explicit ecological and socio-economic objectives as the point of departure for defining metrics to monitor [60], thus are an important step forward. Improving the application of recently developed monitoring guidelines and sharing of lessons learned among practitioners, managers, and academics is particularly important to further the understanding of coral restoration effectiveness.

By surveying projects by various practitioners, the present study for the first time systematically collects information on coral transplantation projects around the world from outside the realm of peer-reviewed scientific literature. However, the information collected was not very detailed and could be refined further. Additionally, as coral reef restoration is a highly dynamic and rapidly evolving field, the present study can only provide a snapshot view. Together with regular monitoring of restoration projects using common metrics, repeated surveys of current

restoration projects are recommended to systematically collate information and make it widely available among a community of interested practitioners and scientists. Communities of practice such as the CRC may serve as a platform for the collection and sharing of information. The questionnaire used in the present study may provide a starting point for future surveys, but should be amended and specified in parts. Additional useful information to be integrated includes: description of specific goals of the restoration project (if any); description of reference sites or conditions (if any); length of time of follow-up monitoring and description of metrics used in monitoring; years of experience of those carrying out restoration; number of species used, amount of transplants per species and year, and rationale for the choice of transplanted coral species; legal/regulatory stumbling blocks or conducive conditions; and in particular technical details such as outplanting techniques and design, as well as experience with and success of different methods. Furthermore, for future surveys it would be highly desirable to obtain a larger number of responses, particularly for regions not covered in proportion to ongoing activity there, such as Australia. This would not only increase representativeness of the results, but also improve statistical robustness.

## Conclusions

While this research focused principally on the method of transplantation, the portfolio of interventions that fit within coral reef restoration strategies is rapidly expanding [9]. Investment in research and development programs such as the Reef Resilience and Adaptation Program (RRAP) in Australia [72] and the Committee on Interventions to Increase the Resilience of Coral Reefs of the National Academies of Science, Engineering and Medicine (NASEM) in the USA [73] are examples of the fast pace at which the field of coral reef restoration is evolving. For these interventions to be successful in the long-term, they will need to be integrated within resilience-based management frameworks rather than on their own as a quick fix [8,9]. For example, reactive interventions on the reef will need to be supported by proactive interventions at local scales (e.g. improved water quality, increased education and awareness) and global scales (e.g. actions on climate change). Our results highlight that there is still a gap between the practice and the science of coral transplantation and restoration. Along with the sense of urgency that fuels the implementation of restoration globally, better communicating the science and guidelines that are being developed is critical to ensuring long-term success rather than "quick fixes" that are likely to fail.

## Supporting information

**S1 Table. Anonymized results of the survey.**
(XLSX)

**S2 Table. Overview of the different projects covered in the survey.** Shown are the country and region in which projects were located, the type of actor running the projects, and the size in terms of amount of corals transplanted. Cases where responses for multiple projects were given by the same respondent are indicated by superscript lowercase letters.
(DOCX)

**S3 Table. Results of text analysis of additional observations offered by survey respondents.** Results are grouped into eight overarching themes, with additional sub-themes and categories as appropriate. Number of comments refers to overall number of times a particular point was mentioned throughout all surveys, and number of sources refers to number of respondents mentioning a particular point.
(XLSX)

**S1 Appendix. Questionnaire used in the survey.**
(DOCX)

## Acknowledgments

We would like to thank the various organizations and individuals who participated in this survey. We are grateful for critical and constructive comments by Alasdair Edwards, James Guest and an anonymous reviewer on earlier drafts of the manuscript.

## Author Contributions

**Conceptualization:** Sebastian C. A. Ferse.

**Data curation:** Sebastian C. A. Ferse, Lena Rölfer.

**Formal analysis:** Sebastian C. A. Ferse, Margaux Y. Hein, Lena Rölfer.

**Funding acquisition:** Sebastian C. A. Ferse.

**Investigation:** Lena Rölfer.

**Methodology:** Sebastian C. A. Ferse, Margaux Y. Hein, Lena Rölfer.

**Project administration:** Sebastian C. A. Ferse.

**Resources:** Sebastian C. A. Ferse.

**Supervision:** Sebastian C. A. Ferse.

**Visualization:** Sebastian C. A. Ferse.

**Writing – original draft:** Sebastian C. A. Ferse, Margaux Y. Hein, Lena Rölfer.

**Writing – review & editing:** Sebastian C. A. Ferse, Margaux Y. Hein, Lena Rölfer.

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
