## [Decision Letter · Decision Letter 0]

24 Dec 2020

PONE-D-20-33086

Current trends and future directions in coral transplantation for reef restoration

PLOS ONE

Dear Dr. Ferse,

Thank you for submitting your manuscript to PLOS ONE. After careful consideration, we feel that it has merit but does not fully meet PLOS ONE’s publication criteria as it currently stands. Therefore, we invite you to submit a revised version of the manuscript that addresses the points raised during the review process.

Your manuscript entitled “*Current trends and future directions in coral transplantation for reef restoration*” has now been assessed by two experts in the field of coral reef restoration. While both Reviewers and I see the importance of the study in advancing knowledge on the work being undertaken by a diverse array of practitioners, there are some major concerns that need to be addressed before the manuscript can be considered for publications. Firstly, some key questions appear absent from the questionnaire that would greatly aid the interpretation of the results. A follow-up questionnaire is highly recommended to provide the additional information noted by both Reviewers. At a minimum, the manuscript needs to have an expanded discussion section to address the points raised by the Reviewers. Secondly, while I appreciate that additional data collection is not always feasible, as noted by Reviewer 1, the study would greatly benefit from information from Australia, given the immense work taking place by diverse practitioners in this region.  Finally, I urge consideration of the tone of the manuscript as noted by Reviewer 1 when conducting the revisions. Consequently, I am recommending major revisions to allow you the opportunity to consider some of the additional data collection suggested by the Reviewers, as well as addressing the tone of the manuscript noted by Reviewer 1. I look forward to seeing a revised version of this manuscript.

We look forward to receiving your revised manuscript.

Kind regards,

Emma F Camp, Ph.D.

Academic Editor

PLOS ONE

'SCAF acknowledges funding from the German Federal Ministry for Education and Research (BMBF, grant number 01LN1303A); https://www.bmbf.de/en/index.html. The funders had no role in study design, data collection and analysis, decision to publish, or preparation of the manuscript.'

We note that one or more of the authors are employed by a commercial company: Marine Ecosystem Restoration (MER) Research and Consulting

4. We note that Figure 1 in your submission contains map images which may be copyrighted.

We require you to either (a) present written permission from the copyright holder to publish this figure specifically under the CC BY 4.0 license, or (b) remove the figure from your submission:

b. If you are unable to obtain permission from the original copyright holder to publish this figure under the CC BY 4.0 license or if the copyright holder’s requirements are incompatible with the CC BY 4.0 license, please either i) remove the figure or ii) supply a replacement figure that complies with the CC BY 4.0 license. Please check copyright information on all replacement figures and update the figure caption with source information. If applicable, please specify in the figure caption text when a figure is similar but not identical to the original image and is therefore for illustrative purposes only.

Reviewers' comments:

Reviewer's Responses to Questions

**Comments to the Author**

1. Is the manuscript technically sound, and do the data support the conclusions?

Reviewer #1: Partly

Reviewer #2: Yes

2. Has the statistical analysis been performed appropriately and rigorously? 

Reviewer #1: Yes

Reviewer #2: Yes

3. Have the authors made all data underlying the findings in their manuscript fully available?

Reviewer #1: Yes

Reviewer #2: Yes

4. Is the manuscript presented in an intelligible fashion and written in standard English?

Reviewer #1: Yes

Reviewer #2: Yes

5. Review Comments to the Author

Reviewer #1: The manuscript titled “Current trends and future directions in coral transplantation for reef restoration” is intended to be a review of global coral transplanting projects to evaluate under-represented restoration projects from around the world as most published literature only comes from projects in the Caribbean. It highlights the concern that there are few venues for presenting data and lessons learned by projects which may not have the means to publish within peer-reviewed literature. Also, this manuscript highlights the common concern of restoration practitioners that long-term monitoring is needed to evaluate the success of restoration activities. Through a survey disseminated to active practitioners, the authors identifies the types of practitioners conducting coral transplantation, their overall objectives, the morphology of coral used in their work, the type of pre- and post-assessment that is conducted, and the source of coral used. This manuscript has many admirable objectives and identifies several interesting trends, but could benefit from some edits, re-writes, and expanded discussion.

The title may not represent actual context of manuscript (i.e., current trends and future direction are a bit misleading as the trends are those represented within the 43 projects outlined in the surveys and the future directions are those perceived as most important by the authors). Title should include “a survey of” or “review”.

Within the first paragraph of the intro, I suggest placing the two sentences addressing anthropogenic stressors next to each other for continuity.

One of the arguments of this manuscript is that coral transplantation is not a useful ecological tool for reefs where natural coral recruitment is successful as first mentioned by Edwards and Clark (1998). The authors of this manuscript argue that few, if any, practitioners even consider coral recruitment when conducting coral transplantation. Based on the status of coral populations, especially in the Caribbean, many local researchers can probably say with much confidence that current coral cover does not support healthy coral recruitment. The time and resources required to conduct coral recruitment studies or modeling prior to starting the lengthy and costly process of coral restoration may prohibit such studies from being conducted. Additionally, the authors stress the need to know the root causes of reef degradation and emphasize that unless the causes are removed, coral restoration will not be successful. This has long been the argument against coral restoration. But as time goes on, coral reefs continue to be faced by the same, as well as additional biological and anthropogenic stressors. So much so that unless actions are taken to preserve some genetic and species diversity, there will be nothing left to restore. Therefore, I would suggest that the authors consider reducing their focus on these potential reasons for not conducting coral transplantation and instead highlight management strategies and conservation techniques that utilize transplantation and other science-based recovery approaches.

Lines 82-89: “Yet, if coral transplantation and restoration in general are gaining interest and popularity as reef management strategies, the methods should be used within a wider set of resilience-based integrated frameworks that simultaneously or pre-emptively address the source of threats and disturbances. Restoration should be the last point of action in a carefully planned management framework. Understanding the cause of coral mortality, the barriers to natural recovery, and the type of repair necessary to initiate recovery are all important considerations that need to be elucidated prior to undertaking transplantation.” These lines raise very interesting discussion topics but are not sufficiently addressed by the authors. What other type of management strategies should incorporate coral transplantation? Please describe some of the integrated frameworks that could address threats and disturbances which may alleviate the need for transplantation. It would benefit the manuscript if the authors could expand on these ideas within the discussion to provide some insight into developing resilience-based integrated frameworks.

Lines 112-119, the authors raise the issue of accountability, an issue that has been mentioned in other reef restoration focused literature (Morrison 2020, Lirman and Schopmeyer 2016, France 2016). For example, “In recent years, reef restoration or coral transplantation projects are increasingly adopted by the dive and tourism industry or as part of corporate social responsibility programs, raising the question to what extent the principles of good practice set forth e.g. by Edwards and Clark and Edwards or lessons earned from the restoration of other ecosystems are implemented in current coral transplantation projects. Specifically, the questions arise whether causes of coral degradation are identified prior to the adoption of coral transplantation, whether there is a bias towards fast-growing (usually branching) species to produce quick results, and whether adequate follow-up monitoring of transplanted corals is carried out.” Unfortunately, I feel the tone of manuscript focuses the attention on the wrong thing. Rather than focusing on if the “principles of good practice….are implemented” by accountable practitioners who have the appropriate experience, the tone of the manuscript seems to focus on raising concerns over certain restoration practices. But, the problem is that the restoration practices of concern are those that are most likely conducted by practitioners without the appropriate experience, without oversight and most likely without accountability. As examples, experienced restoration practitioners are typically well versed in the causes of reef decline (e.g., chronic, acute, anthropogenic, biological, etc) in the areas where they work, or the causes would be known by local stakeholders. The time and effort that goes into restoration would not be wasted by experienced practitioners in an area where unknown circumstances may cause failure. Additionally, fast-growing species may be the first species available for restoration (other slower growing species may not yet be available especially when mariculture is being used), may be the species of choice due to recent losses due to bleaching/disease, or permits may not be available as collecting whole colonies or fragments of massive colonies is more damaging than from branching species. And, in many cases, monitoring, although sometimes not for as long as some would like, is required for restoration activities with any funding or permitting oversight. Therefore, the tone of this manuscript tends to lump all practitioners into one group rather than distinguishing between experienced and novice practitioners. A question within the survey that indicated the length of time that each group had been conducting restoration may have helped with this.

Line 146: I would use the word “focus” rather than “bias” when discussing the use of fast-growing or branching species for coral transplantation. The reasoning behind using fast growing species can be varied and is not necessarily non-altruistic.

Line 147: I would rephrase objective vi to simply state “the source of corals for transplantation”.

Line 182, change “were” to “where”.

It is unfortunate that only 43 programs responded to the survey and the paucity of responses may have skewed the overall outcome of the survey. With the number of list serves, consortiums, conference websites, webinars, and other social media outlets, I am curious what would happen if the survey was presented again. I believe that this is a big setback for the manuscript, especially the lack of projects from Australia, where large interest in active restoration is occurring due to recent losses on the Great Barrier Reef.

Lines 196-197: The authors mention that the distinction between “private” and “business” was not always clear for type of organization, therefore, the wording of the survey may have caused some issues with the results of the survey. For the most part, I believe that the authors handled the categorization of the answers well.

However, in the case of the funding sources, what were the intended differences between Business, NGO and Private?? For example, there are private businesses, NGOs that take private donations and government funding, etc. Without some explanation of the differences between funding types, there may be some discrepancies in the survey results.

Lines 211-212: what is the statistical support for the statement that “The scale of the projects did not differ significantly among the types of actors or regions”?

The “Lessons Learned” paragraph (Lines 313-332) provides some interesting information and highlights the overall objective for many of the practitioners which is successful outplanting. I feel this data could have been more valuable to the objectives and goals of the manuscript had they been incorporated into the questions within the survey.

Lines 335-338: “In this review, we provide an overview of coral transplantation projects from around the world beyond those reported in the scientific literature, including the private sector for which little information exists to date, and identify differences in the design of projects and use of methods among different types of practitioners.” Unfortunately, I don’t find this statement to be true. This manuscript focused on the organization, funding, morphology, substrate, and monitoring type, but did little to advance the science and data coming from such projects. In general, “design of projects” and “use of methods” typically includes information regarding outplanting technique (eg., nails, epoxy, cement, wedging, etc.), number of corals, number of species, outplanting design (eg., arrays, plots, rows, random, etc.), etc. But, this manuscript did more to outline the coral transplantation programs than the coral transplantation itself.

Lines 338- 340: “Being able to draw on the knowledge gained by projects not reported in the scientific literature is indispensable to obtain a better overview of current techniques and to tailor future recommendations.” This statement is so very true and was my hope for this manuscript. Unfortunately, this manuscript did not deliver any new trends or new techniques for coral transplantation. It would have been beneficial if the authors could have identified new information and a means for disseminating ideas and data from the private sector and from smaller projects which are often missing from the scientific literature.

Within the discussion regarding the use of fast-growing, branching species (Lines 371-389), I believe that the authors may miss the mark when interpreting the results of the survey. For example, half of the projects used massive or other morphologies which is considerably higher than what it would have been a decade ago. Techniques for propagating massive species have drastically changed over the last 10 years making it easier to raise them within nurseries and methods for fragmenting have also improved survival for collections in the wild. Many projects begin with fast-growing, branching species to create structure and then transplant massives once other species have become established. To date, most research shows that outplanting massive species is difficult due to high rates of predation. It’s not apples to apples when it comes to transplantation. Also, not all projects are able to collect/propagate all species at once whether it’s due to timing, funding, permitting, etc. it's not for a lack of want for using massive species, the technology and ability to use such species just isn't as advanced as that for branching species which are easier to use for restoration. Therefore, the simplicity of the survey questions may not have allowed the projects to fully explain why only branching corals were used (once again, the age of the project/program would be important to know).

What was the size of the projects that used non-maricultured corals? It's easier for smaller projects to source corals from the wild, but any project looking to outplant on a reef or ecological scale should consider mariculture to avoid detrimental collections from the wild. And it is true that more projects in the Caribbean are probably using nurseries (in or ex situ) as these projects have been operating on longer time scales and the availability of wild corals is much, much lower than on other global reefs.

Discussion of Assessment of Reef Condition: Although I understand the premise behind needing to know the underlying cause of coral mortality (i.e., there is no reason to outplant on a reef if a disease is just going to kill the outplants), sadly, the cause of coral cover loss over time is not usually in question and the solutions to prevent further losses are not simple. The need for baseline surveys prior to restoration is, however, completely relevant (Goergen et al 2020) and the push for better management and water quality standards continues but should not outweigh saving remaining diversity and resilience.

Lines 421-422: “coral transplantation may be a waste of time and resources unless the recipient site fails to recruit juvenile corals.” However, in the broader picture, a site can't recruit juvenile corals if there aren't any adults to contribute to the population.

Lines 423-424: “Furthermore, the absence or minimization of threats is a key attribute of ecosystems that are to be restored – transplantation is not likely to succeed unless chronic stressors are removed.” True, but populations that survive episodic stressors may provide recruitment in the meantime and allow a buildup of resilience while chronic stressors are handled?

Lines 425-427: “An in-depth understanding of the causes of degradation is also key to developing goals and objectives for a restoration project and optimizing site-selection so that the project has a chance to withstand future disturbances.” This is a great point to highlight and there have been some good progress on how to better select sites for multi-species restoration that consider past success and failures.

Lines 453-455: “Several of the projects included measures to address environmental conditions, such as water treatment, fishing regulations or closed areas.” What is meant by water treatment?

Lines 455-459: “However, our survey was unable to ascertain the presence of chronic stressors or the original cause of reef degradation, and we thus cannot assess to what extent root causes of degradation or chronic stressors were adequately recognized and addressed in each case, underlining the need for thorough documentation and explicit statement of goals in restoration projects.” The meaning of this statement is a bit unclear, but perhaps this is more of an issue with the survey rather than a problem with documentation?? This information may be available to the practitioners but may not have been outlined within your questionnaire.

Lines 465-469: “One of the most successful coral transplantation projects at a larger scale (>2ha) was initiated by private Corporate Social Responsibility initiative. Such projects may be at an advantage in terms of available funding (i.e. long-term financial sustainability), local social capital and institutional flexibility (i.e. less ‘red tape’) than government-run initiatives.” How are the authors determining “success” here? Assuming it’s based on the number of accompanying measures that were included in this project, I would hesitate to say that this promises “success” in terms of survival, growth, or habitat and species rehabilitation. In addition, what is the oversight for such projects? What type of accountability is there? It’s easy to look like a project is doing more if not everything is done properly?

Lines 521-523: “Improving the application of recently developed monitoring guidelines and sharing of lessons learned among practitioners, managers, and academics is particularly important to further the understanding of coral restoration effectiveness.” Yes!! It is imperative that the value of consistent monitoring is spread throughout the restoration community.

Reviewer #2: This is a really useful piece of work that will enhance our understanding of real-world coral reef restoration projects. It stands out from other work in that it distinguishes between restoration ecology and ecological restoration, something that many people do not do (notably the most recent large-scale review of coral reef restoration that incorrectly categorises scientific studies as restoration projects). I think the fact that they recognise this distinction and that they have compiled a database of previously unpublished real-world restoration projects is of great value.

I just have a couple of major comments and a few minor comments below:

Main comments:

One problem with questionnaire for me is that the question about overall objectives is that it is grouped into 5 very broad categories, but those categories don’t really tell us anything about success. For example, the question about habitat restoration doesn’t ask what practitioners were trying to restore to. Two questions that would have been really great to ask are a) were there clear a priori goals against which to gauge success (if so, what were they) and b) did you identify a reference site against which to judge success. I guess it is too late to do anything about this now, but perhaps valuable to add a section in the discussion talking about the clear lack of adequate goal setting and the lack of reference sites in most restoration projects.

Research was given as an option for the purpose of the project and was reported as the main objective for 33% of the government projects. If research was the main objective, then I cannot really see how these projects can be considered as restoration projects. I think it is fine to leave these in, but it would be good to discuss this in more detail and ask why research would be a major motivation for an actual restoration project (when the main aim should be conservation, restoration or some socioeconomic goal).

Minor comments:

Line 182: “were” should be where.

Results: Lines 208-210: Were the number of transplants reported the total number of corals transplanted over the entire project, or did you try to get some idea of the number transplanted each year?

Line 343: “widely spread” should be wide-spread

6. PLOS authors have the option to publish the peer review history of their article (what does this mean?). If published, this will include your full peer review and any attached files.

Reviewer #1: No

Reviewer #2: **Yes: **James Guest

---

## [Author Response · Author response to Decision Letter 0]

8 Feb 2021

Editor_1: Some key questions appear absent from the questionnaire that would greatly aid the interpretation of the results. A follow-up questionnaire is highly recommended to provide the additional information noted by both Reviewers. At a minimum, the manuscript needs to have an expanded discussion section to address the points raised by the Reviewers

R: We agree with the reviewer and editor that additional questions would have been desirable. As our intention was to gain a first overview of projects not reported in the scientific literature, we did not include as many details as we should have at hindsight. We feel that a follow-up questionnaire at this point would cause more problems than it would solve, as it is very likely that there would be non-responses and at least some of the contacts and people we were in touch with have changed, posing the risk of issues of consistency. However, we hope that by providing an expanded discussion of the potential shortcomings and the desirability of additional information, together with suggestions for future surveys, we will provide a useful basis for further inquiry.

Editor_2: While I appreciate that additional data collection is not always feasible, as noted by Reviewer 1, the study would greatly benefit from information from Australia, given the immense work taking place by diverse practitioners in this region

R: While we do agree that additional information from Australia would have been desirable and are fully aware of the tremendous developments in that region over the past few years, we feel that following up the survey with additional requests would be problematic (see our response to the first comment), in particular if we would add new respondents at a much later date. The field of reef restoration is highly dynamic and particularly at present is developing rapidly, so that any information gathered in a survey will to some extent be dated again upon publication. Yet, we believe our survey provides a valuable baseline to expand upon and gives a good overview of past and present trends. We would like to note that we had already acknowledged the paucity of information from our survey regarding Australia and the recent developments in that region (lines 352-359 in original submission), but concede this point could be emphasized further. We are now including additional information on programs in Australia, such as the RRAP and the restoration hub supported by the Great Barrier Reef Foundation, as good examples of the rapid development of the field. We also mention some very recent results from reports by the International Coral Reef Initiative highlighting increased interest in restoration from country members- and the likelihood that uptake of restoration will increase substantially in the next decade. 

Editor_3: I urge consideration of the tone of the manuscript as noted by Reviewer 1 when conducting the revisions.

Thank you for this suggestion. We realize our tone may have seemed accusing or prejudiced in some instances, which was not our intention. We have carefully checked the passages indicated by Reviewer 1 as well as the manuscript in general to provide a more neutral, balanced tone. 

Reviewer1_1: The title may not represent actual context of manuscript (i.e., current trends and future direction are a bit misleading as the trends are those represented within the 43 projects outlined in the surveys and the future directions are those perceived as most important by the authors). Title should include “a survey of” or “review”.

R: Thank you for this remark, which we agree with. We have changed the title to “A survey of current trends and suggested future directions in coral transplantation for reef restoration

Reviewer1_2: Within the first paragraph of the intro, I suggest placing the two sentences addressing anthropogenic stressors next to each other for continuity.

R: Unfortunately we did not exactly understand what the reviewer was referring to here. The two sentences addressing anthropogenic stressors (“local sources such as pollution, overfishing and destructive fishing” and “the effects of anthropogenic climate change”) are already placed in subsequent sentences.

Reviewer1_3: One of the arguments of this manuscript is that coral transplantation is not a useful ecological tool for reefs where natural coral recruitment is successful as first mentioned by Edwards and Clark (1998). The authors of this manuscript argue that few, if any, practitioners even consider coral recruitment when conducting coral transplantation. Based on the status of coral populations, especially in the Caribbean, many local researchers can probably say with much confidence that current coral cover does not support healthy coral recruitment. The time and resources required to conduct coral recruitment studies or modeling prior to starting the lengthy and costly process of coral restoration may prohibit such studies from being conducted. 

Additionally, the authors stress the need to know the root causes of reef degradation and emphasize that unless the causes are removed, coral restoration will not be successful. This has long been the argument against coral restoration. But as time goes on, coral reefs continue to be faced by the same, as well as additional biological and anthropogenic stressors. So much so that unless actions are taken to preserve some genetic and species diversity, there will be nothing left to restore. Therefore, I would suggest that the authors consider reducing their focus on these potential reasons for not conducting coral transplantation and instead highlight management strategies and conservation techniques that utilize transplantation and other science-based recovery approaches.

R: Thank you for these important remarks. We agree that surveys of some of the background conditions are time- and resource intensive and in some cases sufficient local knowledge may compensate for a lack of formal data. In any case, incomplete information should not be an excuse for inaction (see also our response to your comment 17 below). It is by no means our intention to contribute to a straw-man argument that discourages any attempts at restoration. Our main point is that transplantation efforts may be for naught if transplants subsequently die for reasons that could have been foreseen. However, there are also no-regrets approaches that could and should be taken to alleviate stress on reefs, as we now acknowledge. We have thus rephrased and amended the section referring to the original Edwards & Clark paper, and also amended our discussion of the assessment of reef conditions, acknowledging the difficulty of conducting full assessments e.g. of recruitment.

Reviewer1_4: These lines raise very interesting discussion topics but are not sufficiently addressed by the authors. What other type of management strategies should incorporate coral transplantation? Please describe some of the integrated frameworks that could address threats and disturbances which may alleviate the need for transplantation. It would benefit the manuscript if the authors could expand on these ideas within the discussion to provide some insight into developing resilience-based integrated frameworks.

R: We have provided additional information on resilience-based frameworks in the subsequent section and in the discussion (in the section on “Accompanying measures” and in the “Conclusions”). We refer to particular accompanying measures such as fisheries regulations, watershed management and afforestation, and frameworks such as Adaptive Resilience-Based Management, but refer the reader to additional literature for specifics in order not to unduly extend the scope of the manuscript. We hope to now provide a good balance between additional information and maintaining a focus on coral transplantation in the present MS.

Reviewer1_5: The authors raise the issue of accountability, an issue that has been mentioned in other reef restoration focused literature (Morrison 2020, Lirman and Schopmeyer 2016, France 2016). For example, “In recent years, reef restoration or coral transplantation projects are increasingly adopted by the dive and tourism industry or as part of corporate social responsibility programs, raising the question to what extent the principles of good practice set forth e.g. by Edwards and Clark and Edwards or lessons earned from the restoration of other ecosystems are implemented in current coral transplantation projects. Specifically, the questions arise whether causes of coral degradation are identified prior to the adoption of coral transplantation, whether there is a bias towards fast-growing (usually branching) species to produce quick results, and whether adequate follow-up monitoring of transplanted corals is carried out.” Unfortunately, I feel the tone of manuscript focuses the attention on the wrong thing. Rather than focusing on if the “principles of good practice….are implemented” by accountable practitioners who have the appropriate experience, the tone of the manuscript seems to focus on raising concerns over certain restoration practices. But, the problem is that the restoration practices of concern are those that are most likely conducted by practitioners without the appropriate experience, without oversight and most likely without accountability. As examples, experienced restoration practitioners are typically well versed in the causes of reef decline (e.g., chronic, acute, anthropogenic, biological, etc) in the areas where they work, or the causes would be known by local stakeholders. The time and effort that goes into restoration would not be wasted by experienced practitioners in an area where unknown circumstances may cause failure. Additionally, fast-growing species may be the first species available for restoration (other slower growing species may not yet be available especially when mariculture is being used), may be the species of choice due to recent losses due to bleaching/disease, or permits may not be available as collecting whole colonies or fragments of massive colonies is more damaging than from branching species. And, in many cases, monitoring, although sometimes not for as long as some would like, is required for restoration activities with any funding or permitting oversight. Therefore, the tone of this manuscript tends to lump all practitioners into one group rather than distinguishing between experienced and novice practitioners. A question within the survey that indicated the length of time that each group had been conducting restoration may have helped with this.

R: Thank you for these important remarks. We fully agree that experienced and accountable practitioners are not likely to waste efforts if they had doubts about the prospects of success. This was part of the rationale behind categorizing respondents into different types of actors, with the assumption (to some extent informed by own observations) that e.g. actors from the tourism industry do not have the necessary experience and are more interested in short-term results than in long-term viability. You are correct to point out that time of experience and accountability (e.g. permitting and reporting requirements) are important factors that would be valuable to include. We have revised our discussion of the actors involved in, and background of, the projects, and have added suggestions on including questions regarding experience and accountability in future surveys. 

We also acknowledge the comment on the availability on fast- vs. slow-growing species and added reasons why fast-growing species may be preferred when discussing the choice of species for transplantation. We have carefully revised and adjusted the tone of the manuscript as suggested in further comments and hope you will agree that we now present a more balanced and nuanced coverage of the topic.

Reviewer1_6: I would use the word “focus” rather than “bias” when discussing the use of fast-growing or branching species for coral transplantation. The reasoning behind using fast growing species can be varied and is not necessarily non-altruistic.

R: Changed to “focus on” as suggested

Reviewer1_7: I would rephrase objective vi to simply state “the source of corals for transplantation”.

R: Changed as suggested

Reviewer1_8: Change “were” to “where”

R: Changed to “where” as suggested

Reviewer1_9: It is unfortunate that only 43 programs responded to the survey and the paucity of responses may have skewed the overall outcome of the survey. With the number of list serves, consortiums, conference websites, webinars, and other social media outlets, I am curious what would happen if the survey was presented again. I believe that this is a big setback for the manuscript, especially the lack of projects from Australia, where large interest in active restoration is occurring due to recent losses on the Great Barrier Reef.

R: We agree that the number of respondents likely would be greater if the survey were to be repeated now, in part because of an increased interest and activity in reef restoration over the past 5 years and particularly with regards to projects in Australia. Note that we had already acknowledged the paucity of information from Australia in our survey and the recent developments in that region (lines 352-359 in original submission). While we feel that a renewed survey would introduce more issues than it would solve (see our response to comment R1_5 above), we have added additional information on recent developments in Australia, and in particular make suggestions for follow-up surveys. 

Reviewer1_10: The authors mention that the distinction between “private” and “business” was not always clear for type of organization, therefore, the wording of the survey may have caused some issues with the results of the survey. For the most part, I believe that the authors handled the categorization of the answers well.

However, in the case of the funding sources, what were the intended differences between Business, NGO and Private?? For example, there are private businesses, NGOs that take private donations and government funding, etc. Without some explanation of the differences between funding types, there may be some discrepancies in the survey results.

R: We appreciate there are overlaps and that distinctions are not always clear cut among those categories. Unlike the question regarding self-characterization, the responses to this particular question were given as multiple options, and so we expect that in case of doubt, several options would have been checked by the respondents. We have slightly modified the description in the methods to clarify that funding sources could be multiple (line 174), and added a qualification when describing the results on sources of funding (lines 234-238).

Reviewer1_11: What is the statistical support for the statement that “The scale of the projects did not differ significantly among the types of actors or regions”?

R: This statement is supported by a Likelihood-ratio test. The results are (G) = 7.8821, X-squared df = 8, p-value = 0.4451; and (G) = 4.0775, X-squared df = 4, p-value = 0.3956. We did not want to include non-significant results in the text, but have now included information on the test used.

Reviewer1_12: The “Lessons Learned” paragraph provides some interesting information and highlights the overall objective for many of the practitioners which is successful outplanting. I feel this data could have been more valuable to the objectives and goals of the manuscript had they been incorporated into the questions within the survey.

R: We agree that the additional information provided by the respondents contains interesting and valuable points, several of which at hindsight would have been worthwhile to include into the questionnaire. Although a formal statistical analysis of this information unfortunately was not possible, we felt it is nonetheless of value to present a form of qualitative summary of these responses to the reader. As both reviewers have remarked on points to include in the survey, and based on the additional responses we received, we have now added a section on suggested items to cover in future surveys at the end of the discussion. 

Reviewer1_13: Unfortunately, I don’t find this statement to be true. This manuscript focused on the organization, funding, morphology, substrate, and monitoring type, but did little to advance the science and data coming from such projects. In general, “design of projects” and “use of methods” typically includes information regarding outplanting technique (eg., nails, epoxy, cement, wedging, etc.), number of corals, number of species, outplanting design (eg., arrays, plots, rows, random, etc.), etc. But, this manuscript did more to outline the coral transplantation programs than the coral transplantation itself.

R: We have adjusted this part to state that we provide information on “differences in the size and objective of projects and accompanying measures among different types of practitioners”. Further in the discussion, we refer to the need for additional details on outplanting technique and design, and provide suggestions for future surveys to address this.

Reviewer1_14: This statement is so very true and was my hope for this manuscript. Unfortunately, this manuscript did not deliver any new trends or new techniques for coral transplantation. It would have been beneficial if the authors could have identified new information and a means for disseminating ideas and data from the private sector and from smaller projects which are often missing from the scientific literature.

R: We are sorry to read the presented information did not fulfil your expectations. It is correct that additional details on outplanting technique and design would be desirable, as stated in the previous comment, and we have taken that on as a recommendation. However, by making the “additional remarks” offered by respondents fully available as supplementary information, we hope that some of the more detailed lessons learned not reported in the main text will be useful to the reader. We have added the following sentence to better describe the scope of the present survey: “The present survey constitutes a first step in that regard, but regular, more detailed surveys among practitioners and sharing of structured information among a community of practice are strongly recommended.” We furthermore identify several means by which information and experiences could be shared among a community of practice, such as the new Coral Restoration Consortium and its website and regular surveys among practitioners.

Reviewer1_15: Within the discussion regarding the use of fast-growing, branching species, I believe that the authors may miss the mark when interpreting the results of the survey. For example, half of the projects used massive or other morphologies which is considerably higher than what it would have been a decade ago. Techniques for propagating massive species have drastically changed over the last 10 years making it easier to raise them within nurseries and methods for fragmenting have also improved survival for collections in the wild. Many projects begin with fast-growing, branching species to create structure and then transplant massives once other species have become established. To date, most research shows that outplanting massive species is difficult due to high rates of predation. It’s not apples to apples when it comes to transplantation. Also, not all projects are able to collect/propagate all species at once whether it’s due to timing, funding, permitting, etc. it's not for a lack of want for using massive species, the technology and ability to use such species just isn't as advanced as that for branching species which are easier to use for restoration. Therefore, the simplicity of the survey questions may not have allowed the projects to fully explain why only branching corals were used (once again, the age of the project/program would be important to know).

R: Thank you for these important remarks. We agree that there may be a multitude of reasons for the choice of particular growth forms for transplantation. We have thus expanded our discussion of the type of corals transplanted, and added remarks on potential shortcomings of the survey questions and suggestions for future surveys. 

Reviewer1_16: What was the size of the projects that used non-maricultured corals? It's easier for smaller projects to source corals from the wild, but any project looking to outplant on a reef or ecological scale should consider mariculture to avoid detrimental collections from the wild. And it is true that more projects in the Caribbean are probably using nurseries (in or ex situ) as these projects have been operating on longer time scales and the availability of wild corals is much, much lower than on other global reefs.

R: Indeed, the use of maricultured corals differed depending on the size of projects – it was much less prevalent in the smallest projects (the test results are given a few lines further down). The use of corals from other sources was similar among projects of different sizes. We have added this information to the results, and specifically mention it in the discussion. We also have included the reviewer’s observation on the particular situation of projects in the Caribbean.

Reviewer1_17: Discussion of Assessment of Reef Condition: Although I understand the premise behind needing to know the underlying cause of coral mortality (i.e., there is no reason to outplant on a reef if a disease is just going to kill the outplants), sadly, the cause of coral cover loss over time is not usually in question and the solutions to prevent further losses are not simple. The need for baseline surveys prior to restoration is, however, completely relevant (Goergen et al 2020) and the push for better management and water quality standards continues but should not outweigh saving remaining diversity and resilience.

R: This is an important point that we agree with – perfect is the enemy of good, and incomplete information or shortcomings in management should not be taken as excuses for not taking any action. We have added your point at the end of this section.

Reviewer1_18: However, in the broader picture, a site can't recruit juvenile corals if there aren't any adults to contribute to the population.

R: True, and in that case transplantation likely is a key component of any restoration endeavour. However, in that case as well, environmental conditions need to be adequate enough to allow for transplants to survive and for any produced larvae to successfully recruit. We have added your point to the discussion.

Reviewer1_19: True, but populations that survive episodic stressors may provide recruitment in the meantime and allow a buildup of resilience while chronic stressors are handled?

R: Good point – we have added this to the discussion.

Reviewer1_20: This is a great point to highlight and there have been some good progress on how to better select sites for multi-species restoration that consider past success and failures.

R: Thank you for this remark. We have added a sentence highlighting that valuable information for the selection of suitable sites can be drawn from information on past successes and failures, and provide a recent reference as example.

Reviewer1_21: What is meant by water treatment?

R: This should be wastewater treatment. We have now revised this part, as the responses we refer to do not specify whether these measures were included as part of the restoration projects or recommended for improved success.

Reviewer1_22: The meaning of this statement is a bit unclear, but perhaps this is more of an issue with the survey rather than a problem with documentation?? This information may be available to the practitioners but may not have been outlined within your questionnaire.

R: We recognize the phrasing here was unclear, and agree that this was an issue with the survey, which did not specifically ask for explicit goals or parameters by which achieving of stated goals might be gauged. We have rephrased this sentence and added recommendations for future surveys.

Reviewer1_23: How are the authors determining “success” here? Assuming it’s based on the number of accompanying measures that were included in this project, I would hesitate to say that this promises “success” in terms of survival, growth, or habitat and species rehabilitation. In addition, what is the oversight for such projects? What type of accountability is there? It’s easy to look like a project is doing more if not everything is done properly?

R: We realize that using the term “success” may have been misleading and missing the mark of what we were trying to argue here. Our point was mostly to direct attention to CSR projects initiated by the private sector. The project in question was noteworthy for its scale and outcome in terms of area covered, cost effectiveness, enhanced coral cover, high species diversity and coral survival – we have removed the term “success”, as the project referred to did not specify particular goals or measures of success. We did not, however, specifically refer to accompanying measures with this example, but simply wanted to provide one noteworthy example of a CSR project. We have now further discussed the prospect of business/NGO projects to include accompanying measures such as such as community engagement, livelihood diversification or environmental awareness campaigns, while also mentioning the aspects of oversight and accountability and potential risks to restoration success if these are missing.

Reviewer1_24: Yes!! It is imperative that the value of consistent monitoring is spread throughout the restoration community.

R: Thank you for this supportive statement. We have now included additional suggestions for regular monitoring of restoration projects using common metrics and regular, systematic surveys of practitioners to collect and share information.

Reviewer2_1: One problem with questionnaire for me is that the question about overall objectives is that it is grouped into 5 very broad categories, but those categories don’t really tell us anything about success. For example, the question about habitat restoration doesn’t ask what practitioners were trying to restore to. Two questions that would have been really great to ask are a) were there clear a priori goals against which to gauge success (if so, what were they) and b) did you identify a reference site against which to judge success. I guess it is too late to do anything about this now, but perhaps valuable to add a section in the discussion talking about the clear lack of adequate goal setting and the lack of reference sites in most restoration projects.

R: Thank you for this important remark. The questionnaire indeed did not ask about specific goals set beforehand, associated measures of success, and means to ascertain whether these were met. We agree that these questions are highly valuable to compare across projects and to assess to which extent specific goals were defined, whether measures to systematically gauge progress and success with regards to stated objectives existed, and ultimately how successful the approaches in different projects were. While we have decided against re-surveying the respondents since we feel there would be risks of non-responses and developments since the initial survey that would make additional data very difficult to analyse (see also responses to comments 5, 9 and 12 of reviewer 1 above), we have now included a paragraph on recommended future surveys and questions to include in the discussion.

Reviewer2_2: Research was given as an option for the purpose of the project and was reported as the main objective for 33% of the government projects. If research was the main objective, then I cannot really see how these projects can be considered as restoration projects. I think it is fine to leave these in, but it would be good to discuss this in more detail and ask why research would be a major motivation for an actual restoration project (when the main aim should be conservation, restoration or some socioeconomic goal).

R: Thank you for this remark. Indeed, some of the surveyed projects may not have been “pure” restoration projects, but note that we also did not specifically exclude those: as we state in our description of the study purpose (lines 156-159), we were aiming to survey projects by a variety of actors, including those but not limited to outside the academic sector, while targeting “particularly ones not reported in the scientific literature”. Listing this option when asking for the purpose of projects was offered to verify whether projects indeed were “pure” restoration, as this was a self-reporting survey and not all respondents were specifically selected and targeted by us. The large majority of projects indeed did not have research as main purpose (47 projects out of 50). Also, note that that question was multiple-choice, and two of the three projects stating research as a main objective listed others in addition. We have now added some discussion of the objectives of surveyed projects in the section “Objectives of transplantation projects” of the discussion.

Reviewer2_3: “were” should be where

R: Changed to “where” as suggested

Reviewer2_4: Were the number of transplants reported the total number of corals transplanted over the entire project, or did you try to get some idea of the number transplanted each year?

R: The questionnaire only assessed how many corals were transplanted in a project overall (see S1 Text); information on transplants per year is not available unfortunately. We agree that this would be useful to know, as it provides additional information on the scale of a project (i.e. 5000 transplants in one go, or over a span of 10 years), and have added a suggestion for inclusion of this point in future surveys.

Reviewer2_5: “widely spread” should be wide-spread

R: Changed to “wide-spread” as suggested

---

## [Decision Letter · Decision Letter 1]

23 Mar 2021

PONE-D-20-33086R1

A survey of current trends and suggested future directions in coral transplantation for reef restoration

PLOS ONE

Dear Dr. Ferse,

Thank you for submitting your manuscript to PLOS ONE. Both of the Reviewer's have recommended that the manuscript is accepted, and they appreciated the effort taken to address their comments. I have suggested minor edits to give you the chance to address the two additional minor comments made by Reviewer 2. The manuscript has been greatly improved by the change in tone and increased discussion and I commend the effort taken to address all points raised. Please address the two minor comments raised and re-submit. I am then happy to progress the manuscript for acceptance.

We look forward to receiving your revised manuscript.

Kind regards,

Emma F Camp, Ph.D.

Academic Editor

PLOS ONE

Journal Requirements:

**Comments to the Author**

1. If the authors have adequately addressed your comments raised in a previous round of review and you feel that this manuscript is now acceptable for publication, you may indicate that here to bypass the “Comments to the Author” section, enter your conflict of interest statement in the “Confidential to Editor” section, and submit your "Accept" recommendation.

Reviewer #1: All comments have been addressed

Reviewer #2: (No Response)

2. Is the manuscript technically sound, and do the data support the conclusions?

Reviewer #1: Yes

Reviewer #2: Yes

3. Has the statistical analysis been performed appropriately and rigorously? 

Reviewer #1: Yes

Reviewer #2: Yes

4. Have the authors made all data underlying the findings in their manuscript fully available?

Reviewer #1: Yes

Reviewer #2: Yes

5. Is the manuscript presented in an intelligible fashion and written in standard English?

Reviewer #1: Yes

Reviewer #2: Yes

6. Review Comments to the Author

Reviewer #1: (No Response)

Reviewer #2: I'm happy that the authors have addressed all of the comments. I think this is a valuable paper and have no hestitation recommending publication. I just have two fairly minor suggestions:

Lines 67-71: It’s not clear to me why this definition for coral reef systems is the correct one to use for coral reefs. Why not just use broadly accepted terms laid out by the SER? Is it useful to basically call any intervention that tries to promote resilience as “restoration”? I wonder if there’s an opportunity to compare and contrast restoration with rehabilitation and make the point that in fact, most reef “restoration” efforts are actually much closer to SERs definition of rehabilitation.

Lines 84-86: I’m not sure about this. Even for sexual propagation, you still have to collect corals from the wild in many cases. These parent colonies can be transplanted back to the reef, but they are often very stressed and die post-transplantation. I would say that the biggest advantages of sexual propagation are increased genetic diversity and access to large numbers of propagules. If gametes are collected in situ then you can also potentially reduce impact on donor reefs.

7. PLOS authors have the option to publish the peer review history of their article (what does this mean?). If published, this will include your full peer review and any attached files.

Reviewer #1: No

Reviewer #2: **Yes: **James Guest

---

## [Author Response · Author response to Decision Letter 1]

25 Mar 2021

Thanks to both reviewers and the editor for their positive assessment of our revisions, we are pleased to see that we have been able to address the comments raised in the previous round of review. We are grateful for the opportunity to further amend the manuscript to address the additional comments by reviewer James Guest. Below we detail how we have dealt with each of the comments.

Reviewer #2_1, lines 67-71: It’s not clear to me why this definition for coral reef systems is the correct one to use for coral reefs. Why not just use broadly accepted terms laid out by the SER? Is it useful to basically call any intervention that tries to promote resilience as “restoration”? I wonder if there’s an opportunity to compare and contrast restoration with rehabilitation and make the point that in fact, most reef “restoration” efforts are actually much closer to SERs definition of rehabilitation.

Response: Thank you for this remark. While the original SER primer acknowledges that in practice, the distinction is often blurred and that “restoration, as broadly conceived [in the primer], probably encompasses a large majority of project work that has previously been identified as rehabilitation” (SER 2004:12), the emphasis of the recent reef-specific definition we have quoted points to an increasing focus on ecosystem services as underlying motivation for reef restoration, and also to the fact that the use of historical baselines (and thus the definition of reference ecosystems, a main attribute of restoration work as per the 2004 SER primer) are increasingly challenging given the rapid changes reef ecosystems are undergoing currently. We have thus amended this section as follows (lines 68-84): “An ecosystem is considered as successfully restored when it features sufficient biotic and abiotic resources to sustain itself structurally and functionally, attaining full recovery when all key ecosystem attributes (including absence of threats, species composition, community structure, physical conditions, ecosystem function, and external exchanges) resemble those of a reference system [8]. While restoration in a stricter sense aims at the re-establishment of pre-existing species composition and community structure, ecosystem rehabilitation emphasizes the reparation of ecosystem processes and services [7]. A recent definition of “restoration” for coral reef systems as “any active intervention that aims to assist the recovery of reef structure, function, and key reef species in the face of rising climate and anthropogenic pressures, therefore promoting reef resilience and the sustainable delivery of reef ecosystem services” [9,p.8] underlines a focus on ecosystem processes and services, and indicates that many reef restoration efforts at present are closer to the Society for Ecological Restoration’s definition of ecosystem rehabilitation rather than restoration [7]. This may reflect the prevalent motivations underlying reef restoration efforts, but is also due to the fact that reef systems are rapidly changing, complicating the use of information on earlier community composition (“historical baselines”) as reference for restoration [9, 10].” 

Reviewer #2_2, lines 84-86: I’m not sure about this. Even for sexual propagation, you still have to collect corals from the wild in many cases. These parent colonies can be transplanted back to the reef, but they are often very stressed and die post-transplantation. I would say that the biggest advantages of sexual propagation are increased genetic diversity and access to large numbers of propagules. If gametes are collected in situ then you can also potentially reduce impact on donor reefs.

Response: That is a valid point – we have modified the sentence to emphasize firstly the advantages in terms of genetic diversity and propagule numbers, and have qualified the potential benefits in terms of reliance on wild-harvested colonies (lines 97-102): “Advances in the propagation and rearing of corals from sexually-produced larvae are helping to increase genetic diversity and provide access to a large number of propagules for eventual outplanting, and in many cases have also reduced the need to obtain corals from the wild as source of transplants, particularly when collection of gametes is done in situ without the temporary removal of corals from the reef [26-28].”

---

## [Editor Report · Decision Letter 2]

29 Mar 2021

A survey of current trends and suggested future directions in coral transplantation for reef restoration

PONE-D-20-33086R2

Dear Dr. Ferse,

Thank you for addressing the minor comments of Reviewer 2. 

We’re pleased to inform you that your manuscript has been judged scientifically suitable for publication and will be formally accepted for publication once it meets all outstanding technical requirements.

Kind regards,

Emma F Camp, Ph.D.

Academic Editor

PLOS ONE

---

## [Editor Report · Acceptance letter]

31 Mar 2021

PONE-D-20-33086R2 

A survey of current trends and suggested future directions in coral transplantation for reef restoration 

Dear Dr. Ferse:

I'm pleased to inform you that your manuscript has been deemed suitable for publication in PLOS ONE. Congratulations! Your manuscript is now with our production department. 

Kind regards, 

on behalf of

Dr. Emma F Camp 

Academic Editor

PLOS ONE